# Response of an Indicator Species, *Dryopteris crassirhizoma*, to Temporal and Spatial Variations in Sika Deer Density

**DOI:** 10.3390/biology11020302

**Published:** 2022-02-12

**Authors:** Yoshihiro Inatomi, Hiroyuki Uno, Mayumi Ueno, Hino Takafumi, Yuichi Osa

**Affiliations:** 1Research Institute of Energy, Environment and Geology, Hokkaido Research Organization, Sapporo 060-0819, Hokkaido, Japan; ueno-mayumi@hro.or.jp (M.U.); hino-takafumi@hro.or.jp (H.T.); osa@hro.or.jp (Y.O.); 2Institute of Agriculture, Tokyo University of Agriculture and Technology, Fuchu 183-8509, Tokyo, Japan; unoh@go.tuat.ac.jp

**Keywords:** wildlife protection area, distance sampling, grazing intensity, line transect, *Cervus nippon yesoensis*, *Sasa nipponica*

## Abstract

**Simple Summary:**

Deer can affect forest ecosystems through foraging behavior. Using indicator species that are sensitive to temporal and spatial variations in deer density has the advantage of managing deer effectively and practically. We examined the response of *Dryopteris*
*crassirhizoma* to the variations in sika deer density in Hokkaido, Japan. We showed that the grazing intensity of *D. crassirhizoma* was sensitive to short-term decreases in deer density and positively related to spatial variation in deer density within regions. *Dryopteris crassirhizoma* can be a useful indicator species and using grazing intensity could help managers rapidly determine their management direction and decide where to focus their efforts.

**Abstract:**

Identifying appropriate indicator species for the impact of deer on forest vegetation is crucial for forest management in deer habitats and is required to be sensitive to temporal and spatial variations in deer density. *Dryopteris crassirhizoma* was selected as a new indicator to evaluate the response to these variations. We examined the population-level characteristics, morphological characteristics at the individual level, and grazing intensity of *D. crassirhizoma* at temporally different deer density sites in Hokkaido, Japan. The response of *D. crassirhizoma* to spatial variation in deer density was also examined within and between two regions in Hokkaido, Japan. Although the population-level characteristics and morphological characteristics did not significantly respond to short-term decreases in deer density, grazing intensity significantly decreased with decreasing deer density. The grazing intensity was also positively related to the spatial variation of deer density within both regions, but the estimated coefficient of the grazing intensity differed between regions. We concluded that *D. crassirhizoma* can be a useful indicator species of the impact of deer on forest vegetation. The grazing intensity of the indicator species was sensitive to temporal and spatial variations in deer density within the region.

## 1. Introduction

Deer can modify the structure and composition of forest plant communities through their foraging behavior [1]. Deer overabundance has various impacts on forest ecosystems, such as declining understory vegetation [2,3], tree debarking [4,5], seedling browsing [6,7,8], declining seed banks of palatable species [9], and soil disturbance [10]. Monitoring these effects as well as deer density is important in developing a deer management plan to reduce these impacts.

The use of plant indicator species has the advantage of reducing the monitoring effort to evaluate deer impacts on forest vegetation [11,12,13]. Previous studies have proposed various indicator species, such as *Trillium grandiflorum* [11,14], *Maianthemum canadense* [15,16,17], *Laportea canadensis* [18], and *Chelone glabra* [19], which are correlated with deer density and grazing intensity. However, these indicators can be rare in heavily affected forests and the sample size to monitor the impacts may be insufficient when they are intolerant to deer herbivory. Deer–vegetation relations can be divided into several phases: only highly preferred plants decline during the first phase, debarking of trees occurs in the second phase, and unpalatable plants dominate in the last phase [8]. The plant species that are selected by deer and are tolerant enough to remain abundant throughout these phases may be a good indicator [8,17]. However, compared with intolerant species, few studies have used a grazing-tolerant species as an indicator of the impact of deer on forest vegetation.

Indicator species are required to be sensitive to temporal and spatial variations in deer density, because detecting these variations can help managers decide where to focus their efforts and rapidly determine the management direction. Population-level characteristics (e.g., coverage and plant density), morphological characteristics at the individual level (e.g., height and leaf length), and grazing intensity of indicator species are often used as indices of the impact of deer on forest vegetation [16,20]. Population-level characteristics may respond slowly to changes in deer density compared with morphological characteristics [17]. However, morphological characteristics such as the height of *Trillium* spp. can be influenced not only by deer usage in the growing season, but also by historical deer usage [21]. The grazing intensity with scars only in the growing season may be more sensitive to temporal changes in deer density than morphological and population-level characteristics. However, few studies have compared the responses of population-level characteristics, morphological characteristics, and grazing intensity of indicator species to temporal changes in deer density. Sika deer (*Cervus nippon*) forage on various plant species. Previous studies have reported that 646 of the 900 species are foraging plants of sika deer in Japan. However, they can drastically shift their foods under conditions of food limitation and there have been cases where the sika deer preference for the same plant species differed in different regions [22,23,24]. If so, we may mislead deer impacts by using indicator species between regions. Thus, it is important to evaluate the utility of the indicator species not only within the region, but also between regions.

*Sasa nipponica* Makino et Shibata is a type of dwarf bamboo and a dominant evergreen understory species in Japan [25]. Because *S. nipponica* is an important forage plant that is grazing-tolerant of sika deer, their height and coverage are often used as indices of deer impact on forest vegetation in Japan [3,7,26,27]. However, it takes a lot of effort to quantify the grazing intensity of *S. nipponica* because the current grazing scars and the previous grazing scars are mixed in the current and wintered leaves of the dense culms. *Dryopteris*
*crassirhizoma* Nakai is a large and perennial semi-evergreen fern that occurs in Japan, eastern Russia, Korea, and northeast China, and often dominates the understory of deciduous broad-leaved forests in northern Japan [28,29]. From May to June, *D. crassirhizoma* grows new leaves, which are arranged in a funnel shape, and the leaves are shed between October and November, except for some overwintering leaves in Hokkaido, Japan [30,31]. New leaves are often grazed by sika deer [22,27,32]. Because *D. crassirhizoma* is larger than other understory species and the grazed new leaves can remain on the forest floor during the growing season, it is easy to find, and the grazing intensity can be measured more easily than that of *S. nipponica*. Thus, this fern can be an appropriate indicator of deer impacts if it is tolerant of herbivory and is sensitive to temporal and spatial variations in deer density. However, no studies have evaluated the response of *D. crassirhizoma* to sika deer density or other environmental and topographic factors.

Estimation of absolute deer density is important when evaluating the relationship between vegetation indices and deer density because the relative deer density may differ temporally and spatially [3]. For instance, deer density may site-specifically change in the short term due to the change in hunting regulations, such as the lifting of hunting bans in the wildlife protected area, and the indicator species may respond to short-term changes. Thus, a site where a hunting regulation has changed should provide a good opportunity to examine the response of indicator species to temporal and spatial variations in deer density.

Our objective was to evaluate the response of population-level characteristics, morphological characteristics, and grazing intensity of indicator species to temporal and spatial variations in sika deer density. To achieve this objective, we selected *S. nipponica* as a proven indicator species and *D. crassirhizoma* as a new indicator candidate, because the coverage and height of *S. nipponica* have been used in previous studies [3,7,26,27], and the plant density, leaf length, and grazing intensity of *D. crassirhizoma* can be measured easily. First, we examined temporal changes in the population-level characteristics (coverage of *S. nipponica* and plant density of *D. crassirhizoma*), morphological characteristics (height of *S. nipponica* and leaf length of *D. crassirhizoma*), and the grazing intensity of *D. crassirhizoma*, and estimated absolute deer density using line-transect methods at the site where the hunting ban was lifted. Second, we examined the response of *D. crassirhizoma* to spatial variation in deer densities within and between two regions with large deer density variations and different deer densities in Hokkaido, Japan.

## 2. Materials and Methods

### 2.1. Study Areas

Our study was conducted in two regions of cool-temperate mixed forests, the Kushiro (KMD) and the Iburi (IMD) districts (Figure 1). KMD covers 144 km^2^ with an average temperature of 5.9 °C, average annual precipitation of 1,329 mm, and a maximum snow depth of 88 cm from 2012 to 2014 at Ota (43°05′ N, 144°47′ E, Figure 1) [33]. The forests consist of coniferous species, such as *Abies sachalinensis* and *Larix kaempferi,* and deciduous broad-leaved species, such as *Fraxinus lanuginosa*, *Tilia japonica*, and *Acer pictum* with an understory of *S. nipponica*, and *Sasamorpha borealis* [34]. Approximately 32 percent of the forest area is planted forest (*A. sachalinensis* and *L. kaempferi*). Sika deer density of KMD was estimated as 15.6 ± 2.3 deer/km^2^ (mean ± SE) in 2014 [34]. In a part of southwestern KMD, there is a wildlife protection area (Figure 1) where deer hunting had been banned from 1964 to 2012 based on Wildlife Protection and Hunting Management Law. However, the ban in part of the area was lifted by the Hokkaido government in October 2012.

IMD, located more than 200 km away from KMD, covers 334 km^2^ with an average annual temperature of 6.6 °C, average annual precipitation of 1,123 mm, and a maximum snow depth of 75 cm from 2012 to 2014 at Hobetsu (42°46′ N, 142°09′ E, Figure 1) [33]. The major vegetation communities are conifer and broad-leaved mixed forests (dominant species are *A. sachalinensis*, *Quercus crispula*, *Tilia maximowicziana*, and *A. pictum*), with understory of *Sasa senanensis* and *S. nipponica* [34]. Approximately 22 percent of the forest area is planted forest (*L. kaempferi* and *A. sachalinensis*). Deer density estimated on IMD was 4.4 ± 0.8 deer/km^2^ (mean ± SE) in 2014, which was lower than that of KMD [34].

### 2.2. Temporal Survey of Indicator Species

In August 2011, two sites were set in KMD. One was a site in the wildlife protection area (WPA), and the other was a site in the neighboring unprotected area (NUA; Figure 1). We established six 10 m × 10 m plots at the WPA and NUA, respectively. These plots were located more than 50 m away from each other. We recorded the diameter at breast height (DBH) using a tape measure for all trees ≥ 1.5 m tall within each plot and determined whether the stem had been debarked by deer. We calculated the ratio of debarked trees to the total number of trees at each site as the debarking rate. The broad-leaved trees (*A. pictum, Acer ukurunduense, Betula ermanii, Betula maximowicziana, F. lanuginose, Hydrangea paniculata, Kalopanax septemlobus, Prunus ssiori, Q. crispula,* and *Sorbus commixta*) and the conifer trees (*A. sachalinensis*) were analyzed together because the ratio of broad-leaved trees to the total number of trees was not significantly different between WPA (53.8%) and NUA (62.4%; *p* = 0.3181, χ^2^ test). We also recorded the number of broad-leaved seedlings 0.5–1.5 m tall (*Actinidia arguta, Celastrus orbiculatus, F. lanuginose, H. paniculate,* and *Lonicera maximowiczii*) within each plot. We did not survey conifer seedlings because they are less palatable to deer [35,36]. On each plot, we randomly arranged one 2 m × 2 m quadrat consisting of four 1 m × 1 m subquadrats on each plot (Figure 2). We took a tree canopy photo using a digital camera (Coolpix 4500, Nikon Corporation, Tokyo, Japan) with fisheye lens (FC-E8, Nikon corporation, Tokyo, Japan) at 1.0 m from soil surface at each subquadrats to estimate canopy openness. Canopy openness was calculated from a photo using CanopOn2 (http://takenaka-akio.org/etc/canopon2/, accessed on 2 November 2021). The mean canopy openness at the subquadrats in the plot was used as the canopy openness of the plot.

In August 2012 and September 2013, the maximum height and coverage of *S. nipponica* were measured in each 1 m × 1 m subquadrat at WPA and NUA. The maximum height of *S. nipponica* was measured once in each subquadrat using steel tape, and the mean maximum height in the subquadrats in each plot was used as the height of the plot. The coverage of *S. nipponica* (categorized as 0, 1, and in increments of 5% for ≥5%) was measured visually by an experienced researcher to prevent observer-dependent bias, and the mean of the subquadrats in each plot was used as the coverage of the plot [37]. We surveyed the coverage and height of *S. nipponica* in one quadrat within each plot because *S. nipponica* is a clonal plant with long crawling rhizomes [25] and they appeared almost evenly within each plot. We recorded the number of grazed and ungrazed *D. crassirhizoma* individuals by deer within each plot at WPA and NUA (Figure 3). We were able to identify grazed *D. crassirhizoma* individuals, because there were no large herbivores other than sika deer in Hokkaido, no livestock grazing was conducted in the forest of the study area, and camera traps were used to check the shape of scars by sika deer. We calculated the ratio of grazed plants to the total number of individuals in a plot as the grazing intensity of the plot. We also measured leaf length of *D. crassirhizoma* individuals as the leaf length from the ground to the tip of the maximum leaf at the plant, and mean leaf length in a plot was used as the leaf length of the plot.

### 2.3. Spatial Survey of Indicator Species

In August 2014, we randomly set ten sites in KMD, including WPA and NUA, and 11 sites in IMD and established two 10 m × 10 m plots at each site (Figure 1). These plots were located more than 50 m away from each other. We recorded the number of grazed and ungrazed individuals and leaf length of *D. crassirhizoma* within each plot. We classified grazed plants into two grazing intensity categories, ’grazed partial-leaves’ and ‘grazed all-leaves’, according to the number of grazed leaves in order to assess the grazing-tolerance of *D. crassirhizoma* (Figure 3). We also recorded the number of herbaceous species and the altitude within each plot. Altitude was measured using a handheld GPS receiver (GPSmap 62SCJ, Garmin Ltd., Olathe, KS, USA).

### 2.4. Line-Transect Survey

A line-transect survey was conducted in our study area from 2012 to 2014. Although the regional density for KMD and IMD during November 2013 and November 2014 were estimated, we additionally estimated site-specific densities within KMD and IMD during the years and for September 2012 [34]. We established one line-transect route at 10 sites in KMD and 11 sites in IMD. The transect route ranged from 2.3 to 5.4 km. The survey was conducted after sunset four times on each route using a vehicle at a speed of 10–20 km/h, and two observers recorded deer group size, the minimum distance between a vehicle and a deer group, and the angle (in degrees) between the direction of travel and the deer group, and the perpendicular distance between the survey line and the group was calculated.

### 2.5. Data Analysis

Temporal changes in the coverage and height of *S. nipponica*, plant density, leaf length, and grazing intensity of *D. crassirhizoma* between WPA and NUA were analyzed using generalized linear models (GLMs). GLM for the plant density was used with a Poisson distribution; GLMs for the coverage, height, and leaf length were used with a Gaussian distribution; and GLM for the grazing intensity was used with a binomial distribution. The explanatory variable included sites in each year (WPA in 2012; WPA in 2013; NUA in 2012; NUA in 2013). Data analysis of the GLM was conducted using R version 4.0.5 [38], and multiple comparisons among the estimated coefficients of the variables were conducted using Tukey’s honest significant difference (HSD) test in the *multicomp* package in R [39].

The relationships between plant density, leaf length, and grazing intensity of *D. crassirhizoma* and spatial variation in deer density, number of herbaceous species and altitude in each region (KMD and IMD) were analyzed using generalized linear mixed models (GLMMs). GLMMs for the plant density were used with a Poisson distribution, GLMMs for the leaf length were used with a Gaussian distribution, and GLMMs for the grazing intensity were used with a binomial distribution. The explanatory variables included mean sika deer density (DD) using the line-transect survey at each site in 2014, number of herbaceous species (NHS), and altitude (ALT) in each plot. Sites were included as random intercepts. We used the Wald chi-squared test to assess the significance of the coefficients. Data analysis of GLMM was conducted using R version 4.0.5 [38] and the *lme4* package [40].

We estimated site-specific sika deer density (DD) for KMD and IMD during 2012–2014 using the conventional distance sampling engine (CDS) with the software, Distance version 6.0 [41]. The CDS engine can select the scale of estimates for density, encounter rate, detection function, and group size using model definition properties [41]. Then, we estimated the density, encounter rate, and group size separately for each site, although the detection function was estimated globally due to the similar landscapes. Before estimating DD, we estimated a detection function, g (x), where x is the perpendicular distance and g (x) is the probability that a deer group is at a perpendicular distance from the line. The data were right-truncated to eliminate 5% of the furthest observations [41]. The truncation distance, *w*, was decided as 120 m. Data were grouped into 10-m intervals of perpendicular distances. The half-normal, hazard rate, and uniform models for the detection function were fitted against the data using cosine, Hermite polynomial, and simple polynomial series expansion terms, sequentially [42]. The selection of the best model and expansion term was based on Akaike’s Information Criterion (AIC) [43]. The density was estimated according to the general equation:DD = n × E (s)/(2 × L × ESW),(1)
where n is the number of deer groups observed, E (s) is the estimated group size, L is the total survey length, and ESW is the effective strip width. The ESW was estimated as the total area under the detection function between 0 and *w* (in m) when g (0) equals 1.0 [42]. Any possible bias in the group size estimation was assessed by the regression of group size (log-transformed) on the estimated detection function. Adjustment for group size was made if the regression was significant at *p* < 0.1 [44]. If not, we used average group sizes. For 2012 and 2013, we used only DDs for two sites, WPA and NUA, because we were interested only in the differences between WPA and NUA. For 2014, we used DDs for all sites in the two regions. The DDs at all sites during 2012–2014 are shown in Table A1.

## 3. Results

### 3.1. Temporal Survey of Indicator Species

The size structure of trees was significantly different between WPA and NUA (*p* < 0.001, χ^2^ test), and there were fewer small trees (≤ 20 cm DBH) in WPA than in NUA (Figure 4). *A. sachalinensis*, *F. lanuginose*, and *P. ssiori* were debarked by sika deer. The debarking rate was 33.8% for WPA and 9.0% for NUA, and there was a significant difference between WPA and NUA (Figure 5; *p* < 0.001, χ^2^ test). The density of broad-leaved seedlings was lower in WPA than in NUA (Table 1; *p* = 0.045, Mann–Whitney U test); however, canopy openness was not significantly different between WPA and NUA (Table 1; *p* = 0.589, Mann–Whitney U test).

The coverage of *S. nipponica* was not significantly different among the sites in each year (Figure 6). The height of *S. nipponica* was lower at WPA than at NUA in both 2012 and 2013, but the difference was not significant (Figure 6). The plant density of *D. crassirhizoma* was lower at WPA than at NUA in both 2012 and 2013, but there was no significant difference between years (Figure 7). The leaf length of *D. crassirhizoma* was not significantly different among the sites in each year (Figure 7). The grazing intensity of *D. crassirhizoma* was higher in WPA than in NUA in both 2012 and 2013, and significantly decreased from 2012 to 2013 (Figure 7). 

### 3.2. Spatial Survey of Indicator Species

The results of the GLMMs showed that the spatial variations in deer density did not significantly affect plant density or leaf length in either KMD or IMD (Table 2). The grazing intensity was positively related to the spatial variation in deer density within both KMD and IMD, but the estimated coefficient of the grazing intensity differed between regions (Table 2 and Figure 8). The number of herbaceous species and altitude positively affected grazing intensity only in KMD but did not significantly affect plant density or leaf length in either KMD or IMD (Table 2). Few individuals lost all leaves by grazing because there were 6 and 16 individuals of grazed all-leaves, and 97 and 137 individuals of grazed partial leaves in KMD and IMD, respectively.

### 3.3. Line Transect Survey

Deer density with 95% confidence interval showed that density for 2012 was significantly higher in the WPA than in the NUA in 2012, whereas deer density for 2013 overlapped between the two sites due to a drastic decrease in WPA (Table 3).

## 4. Discussion

Previous studies have reported that the height of *S. nipponica* decreases with increasing deer density, and the debarking risk for small trees is so high that the number of small trees and broad-leaved seedlings tends to decrease with increasing deer density [3,7,26,45]. Our results on the differences in the height of *S. nipponica*, tree size structure, debarking rate, and density of broad-leaved seedlings between WPA and NUA suggest that the forest vegetation at WPA was heavily affected by sika deer. Although deer hunting was banned at WPA from 1964 to 2012 based on the Wildlife Protection and Hunting Management Law, the ban has been lifted since the 2012/2013 hunting season (from October 2012 to March 2013). We showed that deer density at WPA significantly decreased after the removal of the ban. The ban on hunting for a long time would cause a decline in forest vegetation, and the lifting of the ban might cause a decrease in deer density.

Although almost all *D. crassirhizoma* plants were grazed by deer in 2012, the population-level characteristics (plant density) did not decrease during the following growing season. Augustine and DeCalesta [20] classified tolerance of understory forb species to herbivory into “tolerant”, “partial tolerance”, and “intolerant” according to the leaf area lost by a deer bite and the capacity for regrowth within a growing season. Fern species can be classified in the same way as forbs. Because the body of *D. crassirhizoma* consists of many leaves which are arranged in a funnel shape, it is unlikely that all leaves of the fern are lost by one bite. Indeed, our study showed that few *D. crassirhizoma* individuals lost all leaves by grazing. In addition, *D. crassirhizoma* leaves can regrow after grazing in early spring [46]. Thus, leaves grazed by deer would not lead to immediate withering. Although the percentage of broad-leaved seedlings with browsing incidence can approximate the browsing level [5,37], sufficient seedling data could not be obtained to estimate the browsing level at WPA in contrast to *D. crassirhizoma* data. This suggests that *D. crassirhizoma* is a grazing-tolerant species and is expected to produce a sufficient sample size for monitoring even in heavily affected forests by deer, such as WPA.

Morphological characteristics, such as height and leaf length, may respond almost instantaneously to changes in deer density, compared to population-level characteristics, such as coverage and plant density [17]. In our study, not only the population-level characteristics (coverage of *S. nipponica* and plant density of *D. crassirhizoma*) but also the morphological characteristics (height of *S. nipponica* and leaf length of *D. crassirhizoma*) did not change significantly with decreasing deer density. The height of *Trillium* spp., which is often used as an indicator species, can be influenced not only by deer impact in the growing season, but also by legacy impact [21]. The period of decrease in deer density at WPA may have been too short to detect changes in morphological characteristics. In contrast to the population level and morphological characteristics, grazing intensity of *D. crassirhizoma* significantly decreased with decreasing deer density at WPA. Because the grazing intensity is calculated using only the scars of new leaves in the recent growing season, it can be a sensitive index of short-term changes in deer density.

Previous studies have reported that the grazing intensity of several indicator species correlates with deer density [16,20]. In our study, the grazing intensity of *D. crassirhizoma* in each region increased spatially with increasing deer density. This suggests that grazing intensity can be used as an index of the spatial variation in deer density within a region. However, deer density did not significantly affect plant density or leaf length in KMD and IMD. As mentioned above, because the population-level and morphological characteristics are not sensitive to short-term changes in deer density compared to grazing intensity, it might be difficult to detect the relationship with deer density. In general, plant density and leaf length are expected to be affected not only by deer density but also by various factors, such as climate, soil resources, and the existence of competitive species. In our study, the number of herbaceous species and altitude did not significantly affect plant density and leaf length, but environmental and topographical factors that we did not measure might have influenced them. To appropriately evaluate the relationships between population level, morphological characteristics, and deer density, an analysis considering these factors would be required.

There are cases where sika deer preferences for the same plant species differ between regions [22]. In our study, the estimated coefficient of the grazing intensity to the variation differed between the regions. This suggests that sika deer preferences for *D. crassirhizoma* may differ between regions. Our study showed that the number of herbaceous species positively affected grazing intensity in the KMD. The existence of species that are more attractive than *D. crassirhizoma* might cause a difference in the preference between KMD and IMD. Further studies regarding the effects of peripheral species on deer palatability for indicator species are needed.

Royo et al. [17] recommended including species that are selected by herbivores yet are tolerant enough to remain abundant across the landscape as indicator species. We showed that *D. crassirhizoma* is grazed by deer and is tolerant to herbivory. In addition, we also showed that grazing intensity can be sensitive to temporal and spatial variations in deer density within a region. Furthermore, *D. crassirhizoma* is an easy-to-find species on the forest floor, and the grazing intensity can be measured easily, because it is larger than other understory species and the grazed leaves can remain on the forest floor during the growing season. Therefore, *D. crassirhizoma* can be a useful indicator of the impact of sika deer on forest vegetation in Japan. We propose criteria for selecting indicator species as: (1) deer prefer the species, (2) the species have a tolerance to herbivory, (3) the species can be found easily on the forest floor, and (4) the grazing intensity can be measured easily.

In this study, we demonstrated that the grazing intensity of indicator species is sensitive to temporal changes in deer density, compared with the population level and morphological characteristics. Detecting the short-term impact of grazing intensity could help managers to rapidly evaluate the effectiveness of deer management and determine the management direction. Detecting legacy impact using the population level and morphological characteristics may be suitable for assessing progress toward management goals. For efficient monitoring, we recommend changing the monitoring interval according to the temporal response of these indices. The grazing intensity should be monitored at shorter intervals than the population level and morphological characteristics. We also demonstrated that grazing intensity is sensitive to spatial variation in deer density within the region. Managers could decide where to focus their efforts within their management regions by estimating spatial variation. They should be careful when using the grazing intensity between regions because the differences in palatability for the same species may mislead the evaluation of the impact of deer. In addition to palatability, various factors such as soil resources, site management history, and legacy impacts may cloud the utility of indicator species [16]. To robustly measure the impact of deer on forests, indicator species and their indices (population-level characteristics, morphological characteristics, and grazing intensity) should be selected according to the purpose of the monitoring.

## 5. Conclusions

We conclude that *D. crassirhizoma* can be a useful indicator species of deer impacts on forest vegetation, and the grazing intensity of the indicator species was sensitive to temporal and spatial variations in deer density within the region, compared with the population-level and the morphological characteristics. On the other hands, we should be careful to use the grazing intensity as an indicator to compare between regions because the differences in palatability for the same species may mislead evaluation for deer impacts.

## Figures and Tables

**Figure 1 biology-11-00302-f001:**
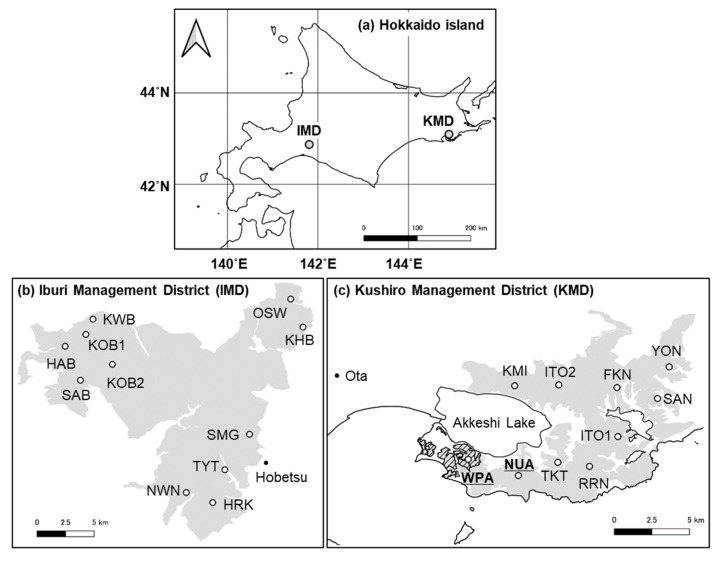
Study regions and sites. Map of (**a**) Hokkaido Island showing the general vicinity of study regions, (**b**) Iburi Management District (IMD), (**c**) Kushiro Management District of Hokkaido Prefectural Forest. The grey areas, the hashed black area, and the letters indicate prefectural forests, a wildlife protection area, and site names, respectively.

**Figure 2 biology-11-00302-f002:**
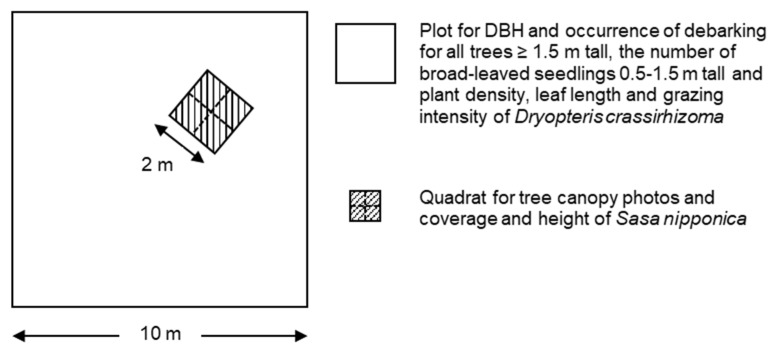
The arrangement of quadrat and subquadrats in each plot. Diameter at breast height (DBH), occurrence of debarking for all trees ≥ 1.5 m tall, and the number of broad-leaved seedlings 0.5–1.5 m tall and plant density, leaf length, and grazing intensity of *Dryopteris crassirhizoma* were recorded within plots. One 2 × 2 m quadrat was arranged in each plot randomly, and tree canopy photos and coverage and height of *Sasa nipponica* were recorded in each 1 × 1 m subquadrat.

**Figure 3 biology-11-00302-f003:**
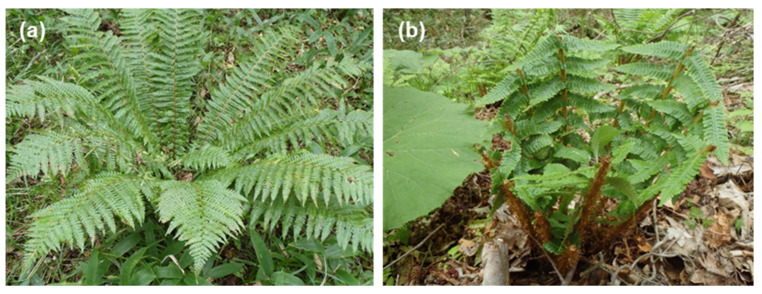
Photographs of *Dryopteris crassirhizoma* individual of (**a**) ungrazed and (**b**) grazed all-leaves by sika deer.

**Figure 4 biology-11-00302-f004:**
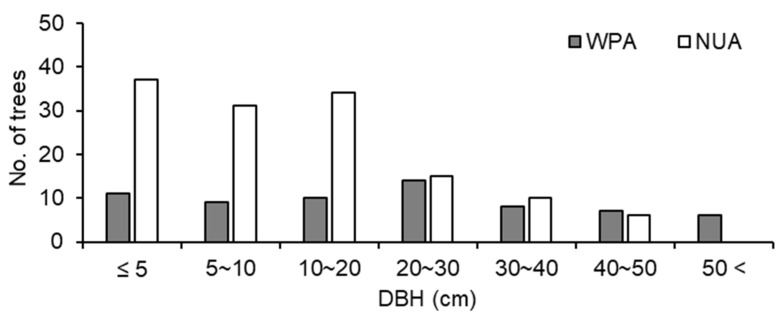
Tree size structure for all trees ≥ 1.5m tall at a wildlife protected area (WPA) and the neighboring unprotected area (NUA) in 2011.

**Figure 5 biology-11-00302-f005:**
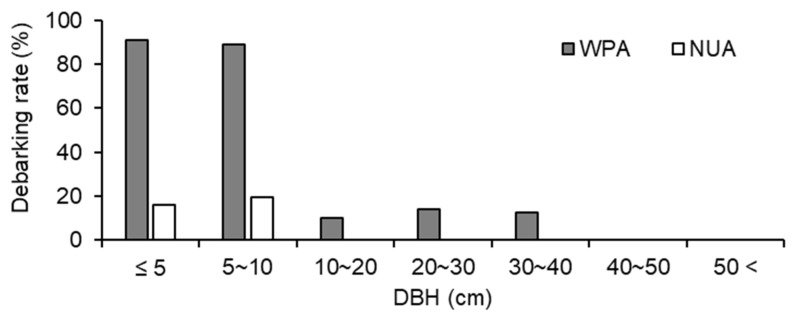
Debarking rates for all trees ≥ 1.5 m tall at a wildlife protected area (WPA) and the neighboring unprotected area (NUA) in 2011.

**Figure 6 biology-11-00302-f006:**
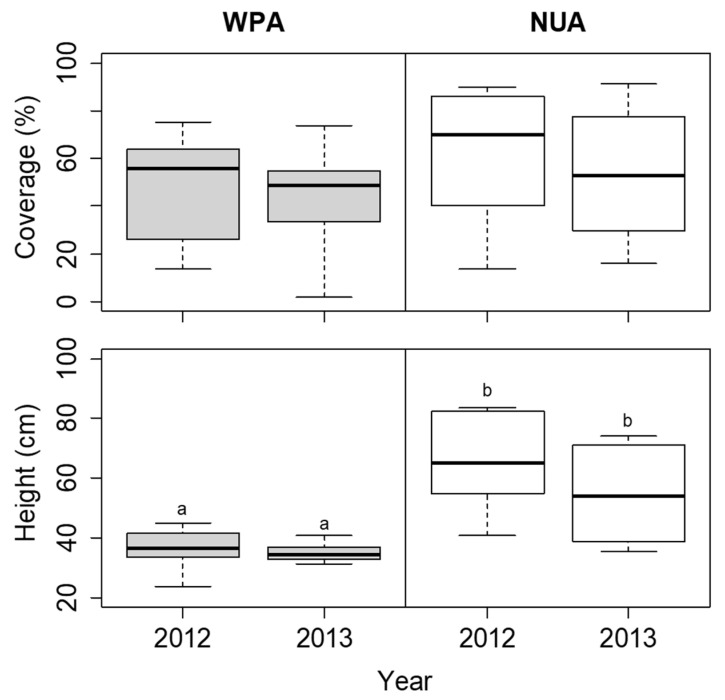
Comparison of coverage and height of *Sasa nipponica* between the wildlife protected area (WPA) and the neighboring unprotected area (NUA) from 2012 to 2013. Deer hunting was banned at WPA until the 2012 survey, but the ban was lifted before the 2013 survey. The boxplots indicate the first quartile, median, and third quartile. Different letters indicate significant difference between sites in each year (α = 0.05).

**Figure 7 biology-11-00302-f007:**
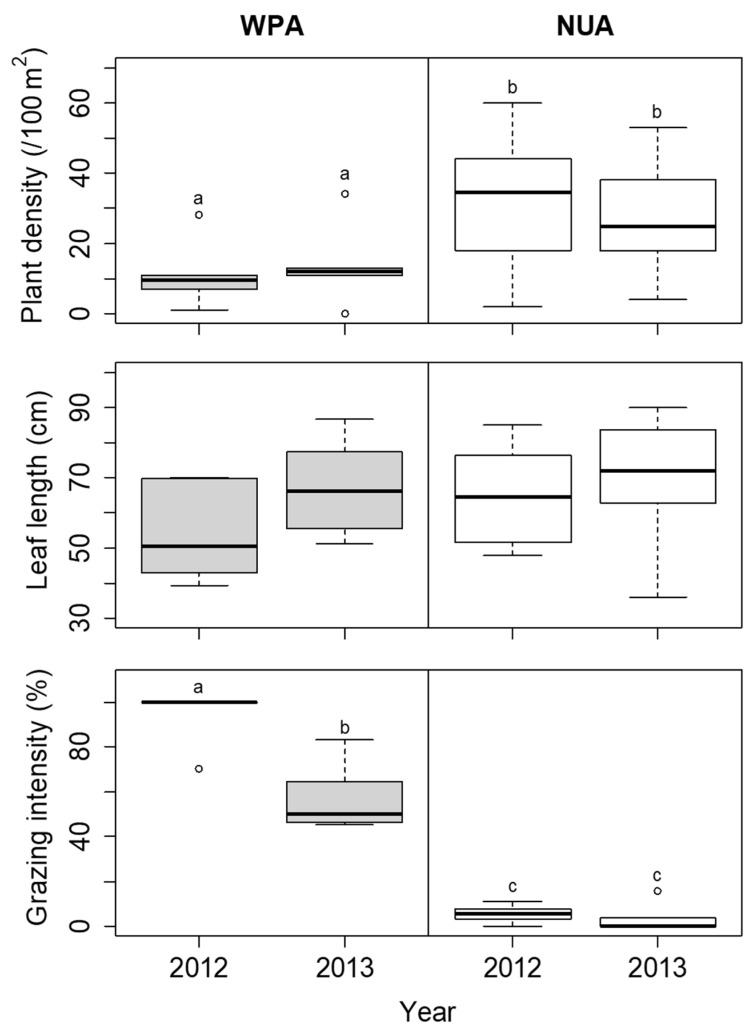
Comparison of plant density, leaf length, and grazing intensity of *Dryopteris crassirhizoma* between the wildlife protected area (WPA) and the neighboring unprotected area (NUA) from 2012 to 2013. Deer hunting was banned at WPA until the 2012 survey, but the ban was lifted before the 2013 survey. The boxplots indicate the first quartile, median, and third quartile. Different letters indicate significant difference between sites in each year (α = 0.05).

**Figure 8 biology-11-00302-f008:**
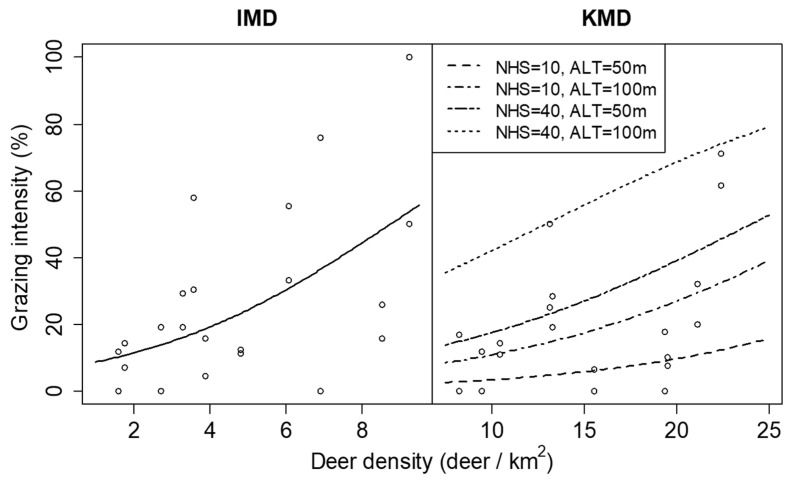
Relationship between grazing intensity of *Dryopteris crassirhizoma* and deer density in Iburi Management District (IMD) and Kushiro management district (KMD) in 2014. Lines are predictions of the generalized linear mixed models and show significant relationships (*p* < 0.05). Lines for KMD represent the cases where the number of herbaceous species (NHS) was 10 or 40 species and altitude (ALT) was 50 or 100 m.

**Table 1 biology-11-00302-t001:** Density of broad-leaved seedlings 0.5–1.5 m tall and canopy openness (±SE) at a wildlife protected area (WPA) and the neighboring unprotected area (NUA) in 2011.

Site	Density of Seedlings (/100 m^2^)	Canopy Openness (%)
WPA	0.33 ± 0.21	8.33 ± 2.13
NUA	5.00 ± 2.59	6.48 ± 1.72

**Table 2 biology-11-00302-t002:** Parameter estimate (± standard error) for the generalized linear mixed models (GLMMs) for plant density, leaf length, and grazing intensity of *Dryopteris crassirhizoma* in Iburi management district (IMD) and Kushiro management district (KMD) in 2014. The explanatory variables included deer density (DD), the number of herbaceous species (NHS), and altitude (ALT).

Response Variables	Explanatory Variables	IMD			KMD		
		Estimate ± SE	χ^2^	*p*	Estimate ± SE	χ^2^	*p*
Plant density	Intercept	3.000 ± 0.450	44.364	<0.001	3.337 ± 0.621	28.857	<0.001
	DD	0.012 ± 0.046	0.072	0.789	−0.026 ± 0.033	0.640	0.424
	NHS	−0.005 ± 0.008	0.3445	0.557	0.009 ± 0.012	0.567	0.452
	ALT	2.619 ± 1.697	2.380	0.123	−2.177 ± 4.333	0.252	0.615
Leaf length	Intercept	83.309 ± 17.434	22.832	<0.001	52.198 ± 20.510	6.477	0.011
	DD	0.277 ± 0.984	0.079	0.778	−0.462 ± 0.788	0.344	0.557
	NHS	−0.300 ± 0.369	0.665	0.415	0.912 ± 0.486	3.525	0.060
	ALT	−32.768 ± 61.780	0.281	0.596	−93.067 ± 174.068	0.286	0.593
Grazing intensity	Intercept	−2.089 ± 1.151	3.291	0.070	−6.250 ± 0.733	72.724	<0.001
	DD	0.257 ± 0.103	6.197	0.013	0.110 ± 0.026	18.055	<0.001
	NHS	0.018 ± 0.021	0.747	0.387	0.059 ± 0.016	12.996	<0.001
	ALT	−5.562 ± 4.812	1.336	0.248	24.539 ± 5.593	19.247	<0.001

**Table 3 biology-11-00302-t003:** Estimated deer density and confidence interval at sites in a wildlife protected area (WPA) and the neighboring unprotected area (NUA) from 2012 to 2013.

Site	Length (km)	Deer Density (Deer/km^2^) (95% Confidence Interval)	
		September 2012	November 2013
WPA	4.0	82.1 (62.1–108.6)	36.5 (31.3–42.5)
NUA	4.5	19.9 (13.3–29.7)	26.2 (21.7–31.6)

## Data Availability

The data presented in this study are available upon request from the corresponding author.

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
