# Peer review of "Response of an Indicator Species, Dryopteris crassirhizoma, to Temporal and Spatial Variations in Sika Deer Density"

_biology, 2022, doi:10.3390/biology11020302_

Round 1
Reviewer 1 Report
the authors describe the use of two species, D. crassirhizoma and S . nipponica, as indicators of sika deer density and grazing effect. The paper is well written and rigorously structured. The work identifies D. crassirhizoma as a good indicator species and the grazing intensity as the best variable associated with the local density variation of the sika deer. In the paper no quantitative data emerges of the availability of D. crassirhizoma in the different study areas and of the presence of other palatable species, which could have influenced the different preferences of sika deer for this species and does not have any data referring to exclusion cages. The work needs an improvement in the description of the methodology referred to the transects for the estimation of the deer in relation to the different sampling units and to the years.
row 147 I suggest better describing the coverage analysis methodology
row 165 it is necessary to better explain how the estimate of deer density per sampling unit was made and in what year
row 170 it is important to describe the lengths of transect for the different vegetation sampling areas
row 217, dds in table 2 to which years do it refer?
row 254 introduce the independent variable in the title
row 257, introduce the year of analysis and statistical significance
row 277 describe the species studied
row 372 specify the species studied and the year
row 375 specify the species studied and the year
Reviewer 2 Report
I carefully read the manuscript entitled “Response of an indicator species, Dryopteris crassirhizoma, to temporal and spatial variations in sika deer density”. The authors evaluated the population-level characteristics, the morphological characteristics at the individual level, and grazing intensity of D. crassirhizoma at temporally different deer density sites and the response of D. crassirhizoma to spatial variation in deer density within and between two regions in Hokkaido, Japan. Also, the authors examined the temporal differences of S. nipponica a known indicator species to deer grazing.
Overall, the manuscript has the potential to provide interesting insights into plant species that could be indicators of increased deer density. However, the ms is not easy to understand and follow by the reader. Also, the sampling design and data analysis look problematic. Methods and results are not clearly structured. Below my general and specific comments on critical points.
General comments
Introduction: authors should make a more detailed literature review and give more details about the sika deer foraging preferences. Also, they have to give more details on the way the indicator species under study respond to sika deer and other environmental and topographic.
Methods: the authors should explain how the study sites were selected (e.g. randomly?). The ms has many methodological problems. Please give information on site differences concerning vegetation and topography. Did the authors statistically tested such differences or examined any other factors concerning the study site that could possibly be responsible for the differences on indicator species density and deer population? Moreover, the authors should identify all the plant species in the quadrats and estimate their cover and structure and use them in an indicator species analysis between sites. Also, this analysis could show other indicator species and perhaps species more preferable by deer. T
The abundance of D. crassirhizoma should be measured in each plot and used as the response variable in GLMM models using deer density, and other topographic (aspect, altitude, slope) and environmental variables (e.g. canopy openness, number of tree trunks in different dbh classes, plant species richness) as explanatory.
Authors estimated the diameter at breast height (DBH), the occurrence of debarking, the number of broad-leaved seedlings, and canopy openness in each plot as background information about deer impacts on forest vegetation. Did these variables differed between sites? Did these variables were related with deer density? Please present these analyses.
Also, the authors do not mention about the other herbivore fauna species of the study area. Are there any other large herbivores? Are there livestock species grazing? How the authors could identify that the leaves were grazed only by the sika deer?
Moreover, the presentation of the methodology is chaotic and no detailed description of the methods and the equipment used is presented. The relative sections should be rewritten in more detail.
Also, the temporal changes in coverage and height should be tested by ANOVA between the years.
Results: the results should be clearer. Please divide the results section into subsections according to the methodology and present the statistics. The results should be rewritten according to the new analysis proposed above. The Appendix give valuable information and I propose to be part of the results.
Discussion: this section is well structured and gives detailed information based on the present form of the ms. However, it should be written according to the new analyses proposed and I willmake more detailed comments after the revision of the ms.
Specific comments
Line 36: substitute “alter” with “modify”
Line 39: please specify the effect on seed banks
Line 41: substitute “indicator species” with “plant indicator species”
Line 47: substitute “Deer-habitat relations” with “Deer-vegetation relations”
Line 139: please specify if 6 plots were established in total or in each area
Line 140-154: please rewrite the session and give more details on the sampling of the environmental variables. E.g. the DBH was measured by equipment or visually? On what kind of trees? Was there a classification of broadleaved and coniferous species or they pulled together and why? How the author calculated the occurrence of debarking? Why the seedlings of coniferous species were excluded? How the authors calculated the canopy openness in each plot? Also, the authors say that one 2x2 m was arranged in each plot. This quadrat was measured in the center of each plot or it was randomly? Why the authors used only one quadrat to measure the indicator species S. nipponica and they did not use more quadrat in order to cover largest area in the sampling plot (I think that using 9 quadrats allocated on a distance of 2 meters between then would be more appropriate to cover the sampling plot). Moreover, the authors estimated visually the cover of the species but they don’t write about the calculation of the maximum height of the species, did they used a meter? How many measures were taken in each subquadrat? Finally, I believe that to measure the grazing intensity of the study indicator species D.crassirhizoma the authors should have taken more sampling quadrats in each plot as mentioned above. Also, the author could recognize that the leaves of the species were eaten by deer and they were not cut off by other herbivores or causes?
Line 156-164: please rewrite the session with more details on the methodology. I have the same questions as above.
Line 166-174: please give more details on the methodology. How many transects were established? Where did they establish? What was the length of each transect? How many times they were visited on each year? What time did the authors visited the transects? What was the speed of the vehicle?
Line 219-230: Please present the significant statistics not presented in the figures and tables
Line 244: please present the standard deviation and the statistics in the table 1
Reviewer 3 Report
Dear Authors,
I send you my comments on the paper ‘Response of an indicator species, Dryopteris crassirhizoma, to temporal and spatial variations in sika deer density’
This is a very interesting and well-structured study, carried out with a simple but effective experimental design.
I find very interesting the approach adopted by the authors. They studied the short-term impact of grazing intensity by sika deer on an indicator fern species. From a methodological point pod view, they selected two different study areas and considered temporal variations in their sampling surveys.
They used GLM for data analysis.
Finally, based on their results, they propose criteria for selecting indicators species.
Moreover, the paper is well written and clearly stated and it is suitable for the Special Issue ‘The Interaction between Large Ungulates and Plants.
I think that it is suitable for publication in ‘Biology’.
Some minor comments:
- Introduction. It could be interesting to state already at the end of the introduction that you selected two study areas with different pressure of deer density.
- the paper reports the results of sampling surveys carried out in 2012 and 2014. Since we are already in 2022, please provide somewhere (M&M ?) a justification for using these ‘not-recent’ data…
- discussion. In my opinion, it is not necessary to repeat the reference to each figure and table in this section.
- it could be interesting to report a figure with some photos of the proposed indicator fern Dryopteris crassirhizoma, just to have a look at its shape, and, if it is possible also examples of grazed leaves.
Round 2
Reviewer 2 Report
The authors adequately addressed my previous concerns. I have no further comments.